# Liver Transplantation for Hepatocellular Carcinoma after Downstaging or Bridging Therapy with Immune Checkpoint Inhibitors

**DOI:** 10.3390/cancers13246307

**Published:** 2021-12-15

**Authors:** Qimeng Gao, Imran J. Anwar, Nader Abraham, Andrew S. Barbas

**Affiliations:** Department of Surgery, Duke University Medical Center, Durham, NC 27705, USA; qimeng.gao1@duke.edu (Q.G.); imran.anwar@duke.edu (I.J.A.); nader.abraham@duke.edu (N.A.)

**Keywords:** hepatocellular carcinoma, immune checkpoint inhibitor, liver transplantation, bridging, downstaging

## Abstract

**Simple Summary:**

Immune checkpoint inhibitors (ICI) have revolutionized the treatment of hepatocellular carcinoma (HCC). In addition to their role in advanced HCC, there is considerable interest in using ICIs in the neoadjuvant setting, either as a downstaging or bridging therapy, prior to potentially curative liver transplantation. In this article, we reviewed all the available literature on ICI use in this context. We postulate that ICIs may be utilized safely prior to liver transplant; however, further research is needed in this area.

**Abstract:**

Liver transplantation offers excellent outcomes for patients with HCC. For those who initially present within the Milan criteria, bridging therapy is essential to control disease while awaiting liver transplant. For those who present beyond the Milan criteria, a liver transplant may still be considered following successful downstaging. Since the introduction of atezolizumab as part of the first-line treatment for HCC in 2020, there has been increasing interest in the use of ICIs as bridging or downstaging therapies prior to liver transplant. A total of six case reports/series have been published on this topic, with mixed outcomes. Overall, liver transplantation can be performed safely following prolonged ICI use, though ICIs may increase the risk of fulminant acute rejection early in the post-operative period. A minimal washout period between the last dose of ICI and liver transplantation has been identified as an important factor predicting transplant outcomes; however, further research is needed.

## 1. Introduction

Liver cancer is the sixth most common cancer worldwide, with hepatocellular carcinoma (HCC) accounting for 90% of cases [1]. Surgical resection and liver transplantation have been the cornerstones of early-stage HCC treatment. The decision to resect or transplant is influenced by many factors, such as tumor characteristics, the presence or absence of portal hypertension, the severity of underlying liver dysfunction, and patient performance status, thus remaining a topic of debate [2]. In general, hepatic resection is preferred among HCC patients with solitary lesions in whom perioperative hepatic decompensation is unlikely, whereas liver transplantation is the treatment of choice among those with multi-focal disease and more advanced liver dysfunction.

The Milan criteria are the primary selection criteria when evaluating the HCC tumor burden in patients being considered for liver transplant [3]. For patients with tumors within the Milan criteria, liver transplantation offers excellent short-term and long-term outcomes, with a 5-year overall survival over 70% and recurrence rate around 10–15% [4,5]. With careful patient selection, similar short-term outcomes can be achieved with tumor resection; however, the risk of recurrence at five years approaches 70% [6]. Furthermore, 10-year survival after liver resection for HCC is as low as 15% [7].

For patients with tumors beyond the Milan criteria, or beyond the intermediate stage based on the Barcelona Clinic Liver Cancer (BCLC) system, locoregional and systemic therapy have become the mainstay treatment options. The last decade has witnessed significant advancements in HCC systemic therapy. Sorafenib, approved in 2008, had previously been the only first-line treatment for advanced HCC [8]. In 2020, atezolizumab, a PD-L1 monoclonal antibody, plus bevacizumab, an anti-VEGF monoclonal antibody, have been shown to improve progression-free survival compared to sorafenib [9]. More importantly, this combination induced a complete response in 5.5% patients compared to 0% in the sorafenib arm of the study. The improvement in systemic therapy for HCC opens the unique possibility of treating HCC in a neoadjuvant context before liver transplantation.

In this review, we present the available literature on the clinical experience with ICIs in the neoadjuvant setting, either as a bridging or downstaging therapy for HCC before liver transplantation. Although there is a paucity of literature in this area, checkpoint inhibitors have been utilized safely prior to transplant. This review highlights potential directions for future research regarding the use of ICIs in HCC treatment.

## 2. Current Bridging and Downstaging Strategies Prior to Liver Transplantation

HCC is unique in that 80–90% cases develop in patients with underlying liver disease. Conceptually, liver transplant represents the ideal therapy, as both the primary tumor is resected and the risk for developing future liver neoplasms is significantly reduced. Unfortunately, the ongoing scarcity of donor organs has limited the utilization of transplantation for HCC to candidates with early-stage disease in an effort to avoid futility [10]. However, since the inception of the Milan criteria, efforts to refine and safely expand selection criteria have been developed and implemented in the field of transplant oncology [11,12,13,14,15,16,17]. As the listing criteria for transplants become less restrictive, the risk of tumor recurrence inevitably increases. For example, the Extended Toronto Criteria offer liver transplantation for patients with any number of tumors of all sizes, barring any extrahepatic disease, vascular invasion, or poorly differentiated status on biopsy. Even though the overall 10-year survival was not statistically different compared to those within the Milan criteria, the recurrence rate after transplantation in this population was close to 30% [16].

On a practical level, the shortage of suitable donor livers remains a major limitation to the field of transplantation. In the US transplant system, in which patients must meet Milan criteria to qualify for MELD exception points, many patients spend months to years on the waitlist. In the US, the median waiting time for candidates with MELD scores 15–34 (the vast majority of HCC patients fall within this category) was 5.6 months in 2019. Moreover, all HCC patients must now demonstrate disease stability for 6 months prior to receiving initial exception points [18]. As such, the transplant rate for patients with HCC exception points is over 50% lower than it was 10 years ago [19]. Within this framework, waitlist mortality and dropout rate are substantial. Of all patients listed for liver transplant in 2016, 11.3% died on the waitlist and an additional 23.3% were removed from the list over a 3-year period. Left untreated, the risk of dropout may approach 25% at 1 year [20].

Bridging therapies, designed to control or delay disease progression, are key for patients on the waitlist. Locoregional therapy (LRT) is the most common form of bridging therapy. LRT is an umbrella term that encompasses a range of therapies, such as radiofrequency ablation (RFA), microwave ablation (MWA), transarterial chemoembolization (TACE), radioembolization (Y-90), and stereotactic body radiotherapy [21]. There have been no randomized trials conducted evaluating the efficacy of LRT as a bridging therapy, and most retrospective comparative studies are subject to bias. Those selected for LRT are more likely to have more advanced tumors. As a result, among those who undergo liver transplantation, LRT bridge therapy has not been shown to improve overall survival or reduce recurrence risk [22,23]. However, because the dropout rate from the transplant waitlist can be reduced by LRT bridge therapy [24], LRT is now accepted as the standard of care for those expected to remain on waitlist for more than 6 months [25].

In addition to treating patients on the waitlist with HCC within the Milan criteria, LRT can be utilized as a downstaging therapy for patients initially presenting with HCC beyond the Milan criteria. For patients who are otherwise not eligible for a potentially curative liver transplant, the ability to effectively downstage patients confers a definite advantage regarding access to transplantation. Many retrospective studies have showed that HCC can be downstaged to within the Milan Criteria with LRT, and if downstaging is successful, long-term recurrence free survival and overall survival following transplant are comparable to those within the Milan Criteria [4,26,27,28,29]. The 2018 American Association for the Study of Liver Diseases (AASLD) guidelines now recommend the consideration of patients beyond the Milan criteria for transplant after successful downstaging to within the Milan criteria [30]. In 2017, UNOS/OPTN adopted an expanded set of inclusion criteria to allow the prioritization of those who are successfully downstaged [4]. The first prospective randomized trial to evaluate the benefit of liver transplantation after successful downstaging with LRT to within the Milan criteria was published in 2020. Out of the 74 patients enrolled, 61% were successfully downstaged and randomized. The 5-year tumor-free survival and overall survival were 76.8% and 77.5% for those who received liver transplantation, compared to 18.3% and 31.2% respectively for those who received non-transplantation therapies [31].

## 3. Immune Checkpoint Inhibitors for HCC

Immune checkpoint inhibitors (ICIs) have revolutionized the field of oncology in recent years. Their success has been replicated for multiple types of cancers, and there are currently numerous ongoing clinical trials designed to evaluate this new class of cancer therapy in different settings. ICIs are particularly attractive for HCC treatment for several reasons: (1) HCC responds poorly to traditional cytotoxic chemotherapies; (2) HCC evolves in the context of underlying chronic liver inflammation, which can lead to exhaustion and the immune escape of tumor cells; (3) the liver harbors a large repertoire of resident immune cells; (4) the liver is considered relatively immuno-privileged and thus tumor immunity is suppressed to some extent. Although there are many checkpoint receptor targets, such as CTLA-4, PD-1, TIM-3, VISTA, and OX-40, most therapies in clinical trials or in current use target CTLA-4, PD-1, and PD-L1. Various agents have been tested in HCC patients in a range of settings.

For early-stage HCC (BCLC stage 0 and A), tumor resection and ablation are the standard of care. Although no ICI treatment combinations have been approved, many are being tested in phase I–III trials in the adjuvant setting [32]. For intermediate-stage HCC (BCLC stage B), the standard of care is TACE or Y90. Although studies that have combined sorafenib and TACE over the years have not shown any added benefits [33,34], there is considerable interest in exploring the potentially synergistic effect between LRT and ICI. Theoretically, transarterial therapies lead to tumor necrosis and the release of tumor antigens, which, in the presence of checkpoint inhibition, can boost antitumor immunity. Additionally, studies demonstrate that TACE may enhance PD-1 expression on intratumor inflammatory cells and PD-L1 expression on HCC tumor cells [35].

For patients with advanced HCC (BCLC stage C) or those with intermediate HCC (BCLC stage B) who either are not a candidate for LRT or who have progressed despite therapy, treatment options are limited. sorafenib was the only first-line therapy approved for HCC until 2018, when lenvatinib, another tyrosine kinase inhibitor, was approved following a phase III non inferiority trial [36]. Most clinical trials on ICIs in HCC have focused on this group of patients. The IMbrave150 trial compared the atezolizumab and bevacizumab combination in advanced HCC patients to standard-of-care sorafenib and found better progression free survival and overall survival among those who received the atezolizumab-bevacizumab combination [9]. This ultimately led to the approval of atezolizumab-bevacizumab as a first-line treatment for advanced HCC. Similarly, three additional ICI therapies—nivolumab monotherapy, pembrolizumab monotherapy and nivolumab/ipilimumab dual therapy—have been approved by the FDA as second-line therapies based on published phase I–II trial results [37,38,39]. Furthermore, a number of ongoing trials are evaluating other ICI agents either as monotherapies or in combination with other treatment strategies [40].

Thus far, the toxicity of ICI agents appears to be limited. ICI-induced hepatoxicity (ICH) is a distinct entity and often presents as asymptomatic elevation of liver enzymes [41]. The median time to onset is around 10 weeks but can manifest as early as 2–3 weeks following the initiation of treatment. Fulminant liver failure can result but is rare. The underlying mechanism is not fully understood but may be multi-factorial, including the cross-reactivity of anti-tumor antibodies against hepatocytes, excessive cytokine production, and the development of autoantibodies [42]. Based on large clinical trials, ICIs are well tolerated among patients with underlying liver disease and the rate of ICH is comparable to that of patients without liver disease. However, these trials included only patients with Child–Pugh class A and B (7–9) cirrhosis [9,37,43].

## 4. Downstaging HCC with Neoadjuvant Immune Checkpoint Inhibitors

The use of ICIs as bridging or downstaging therapies prior to liver transplant has been described, although experience remains quite limited. A major concern limiting ICI use is the potential to increase the risk of acute rejection and lead to graft loss. In the post-transplant setting, studies on ICI use have shown that a significant portion of transplant recipients may experience biopsy-proven acute graft rejection when receiving ICIs for various malignancies. A recent review by Yin et al. [44] summarized outcomes on 28 liver transplant recipients who received ICIs. Eighteen received ICIs for HCC recurrence and the rest for melanoma and other types of cancers. Thirty-two percent of those patients experienced rejection. Similar rates of acute rejection were observed among kidney transplant recipients receiving ICIs, with over half of those that developed rejection losing their allograft [45].

To date, only six case reports have described ICI use either as a bridging or downstaging therapy for HCC prior to transplant [46,47,48,49,50,51]. All are single-center studies—two in the United States, two in Europe, and two in China. The 20 patients from these studies are presented in aggregate (Table 1). Most patients were male (85%) and had viral hepatitis as the cause of underlying liver disease. All except one received other forms of treatment, including surgical resection, sorafenib and LRT (information available for 13 patients only). Most patients (77%) were within the Milan criteria at the time of transplantation, whereas the rest were within the UCSF criteria. A range of ICIs were used as monotherapies, but all targeted the PD-1/PD-L1 pathway. Fifty-five percent of patients received nivolumab [46,47,51], whereas the rest received pembrolizumab, camrelizumab, toripalimab, or durvalumab [48,49,50]. The duration of ICI treatment varied from 6 weeks to over 2 years. The washout period between the last dose of ICI and liver transplant ranged from 1 day to 253 days.

Outcomes varied significantly among studies. Four studies reported successful transplant after ICI use [47,49,50,51]. Two patients (out of 18) had mild rejection that was successfully treated with the adjustment of their immunosuppression regimens. Poor outcomes were described in two studies [46,48], with both patients dying from liver failure early in the post-op course. Nordness et al. [46] described a patient with treated HCV who developed HCC recurrence following initial liver resection. His tumor burden was within the Milan criteria following sorafenib and Y90 treatment; however, he was not listed due to elevated AFP. Nivolumab was started and promptly controlled his disease progression. After a year without disease progression and remaining within the Milan criteria, he was listed for liver transplantation. Nivolumab was continued as a bridging therapy and the patient received a liver transplant from an HCV-antibody-positive NAT-negative donor, 8 days after his last dose of nivolumab. A donor liver biopsy showed no evidence of hepatitis, fibrosis, necrosis, or other abnormalities. Postoperatively, the liver enzymes failed to normalize, despite therapeutic tacrolimus levels and good vascular flow demonstrated on Doppler ultrasound and contrasted CT imaging. A biopsy on postoperative day 6 showed acute hepatic necrosis with dense lymphocyte infiltration. Despite salvage high-dose steroids and rabbit anti-thymocyte globulin (ATG) treatment, the patient continued to deteriorate and expired on day 10. Chen et al. [48] reported a similar case. Following recurrence after initial surgical resection, the patient received lenvatinib and toripalimab as a bridging therapy. The last dose of toripalimab was 93 days prior to the liver transplant. Despite an uncomplicated surgery, the patient did not regain liver function and eventually progressed to multi-organ failure and death on postoperative day 3. Liver biopsy on postoperative day 2 indicated massive hepatic necrosis. It is unclear whether ICI use was the culprit in this case, as there was no obvious evidence of rejection on pathology. The authors agreed that this could have represented primary graft nonfunction.

The washout period between the last dose of ICI and liver transplantation is clearly an important factor to consider. Schwacha-Eipper et al. discontinued nivolumab for 6 weeks before activating the patient on the waitlist [47]. A similar strategy was employed by Sogbe et al., (3 months) [50]. However, there is insufficient evidence to support a minimal washout period. In their nine-patient cohort, Tabrizian et al. reported successful transplantations 1 day and 2 days after the last nivolumab infusion [51]. It is important to note that both patients had significant blood loss intraoperatively, prompting 30-unit and 15-unit red blood cell transfusions, respectively. The blood product replacement potentially accelerated the washout of ICIs. Nevertheless, cases of successful transplant within 4 weeks of their last ICI dose have been described [49,51]. Additionally, the minimal washout time was often set loosely based on the reported serum half-life for ICIs (Table 2); however, the occupancy of the ICI pharmacological target can remain high for significantly longer. For example, the serum half-life for nivolumab is 12–20 days, but a sustained occupancy of over 70% of PD-1 molecules on circulating T is are observed more than 2 months following a single infusion. In some patients, PD-1 occupancy can remain above 50% for 200 days after the last dose following repeated infusion [52,53]. This prolonged effect of ICIs is supported by the often-delayed onset of side effects among patients receiving ICIs [54]. Overall, it is possible that a short interval between ICI infusion and transplant increases the risk of acute rejection post-operatively, but the quality of evidence has been limited thus far. Serum half-life of ICIs may be a poor indicator when deciding the minimal washout period, given the long-lasting effect on target receptors. Furthermore, since ICI binding to their corresponding target is reversible, it may be worthwhile to consider modalities such as plasmapheresis in order to accelerate washout if needed. A minimal washout period, however, could be useful in a different context, to potentially stratify patients. Those who progress early, especially systemically, upon discontinuation of ICIs, may not derive a benefit from a liver transplant.

The PD-1/PD-L1 pathway is a key negative regulator of the immune response and is involved in organ acceptance after transplant [65]. Studies on transplant recipients receiving ICIs suggest a correlation of graft PD-L1 expression with risk of graft rejection following ICI [66,67,68]. Of patients whose graft biopsy stained negative for PD-L1, none experienced acute rejection following nivolumab or pembrolizumab infusion, whereas those whose allograft stained positive for PD-L1 all experienced acute rejection [67,68]. Therefore, PD-L1 expression on liver allografts could theoretically serve as a potential marker in the context of anti-PD-1 ICI use. Its utility during the peri-transplant period, however, remains unknown. Both Nordness et al. and Chen et al. reported graft PD-L1 expression on immunohistochemistry—negative staining on pre-implant biopsy but positive on post-operative biopsies [46,47,48]. However, PD-L1 expression correlates with immune activation and could be a result of acute rejection [69]. Lastly, the use of anti-CTLA4 therapy in the neoadjuvant setting prior to liver transplantation has not been reported in the literature. Some case reports on ipilimumab use in the transplant patient population suggest that targeting CTLA4 may be associated with a lower risk of rejection [70,71].

Although not the primary focus of this review, neoadjuvant ICI has been utilized prior to definitive liver resection as well. The primary aim of neoadjuvant ICI in this context is to achieve downstaging followed by resection for those initially presenting with unresectable disease. However, this approach also has the potential to reduce the risk of recurrence for those presenting with initially resectable tumors. In the first scenario (downstaging of unresectable tumors), two case studies to date have described the neoadjuvant use of TACE with tislelizumab or camrelizumab in HCC patients before resection [72,73]. Chao et al. [72] reported on a patient with underlying cirrhosis presenting with a large tumor involving both right and middle hepatic veins, without extrahepatic metastasis. Following two rounds of TACE and tislelizumab, imaging showed a complete radiological response. The tumor was then resected safely and there was no disease recurrence at 6-month follow-up. Xin et al. [73] described two similar cases. The first patient presented with a large liver primary tumor, associated satellite lesions and a metastatic lesion in the spleen and the second patient presented with a bulky tumor with left and middle hepatic vein invasion and IVC tumor thrombosis. Both patients received TACE, neoadjuvant camrelizumab, and subsequent resection. Post-resection, both patients received adjuvant camrelizumab and had no recurrence on a 6-month follow-up. Given the small sample size and short follow-up, it remains unclear whether ICI should be utilized routinely before salvage resection. More clinical trials (NCT03510871 and NCT03299946) are currently in process to evaluate the efficacy of ICIs in borderline-resectable or unresectable HCC [74]. In the second scenario (the use of ICIs to reduce recurrence post-resection), early trial results have been encouraging, with a 37.5% pathological complete response observed in the interim analysis [75,76].

## 5. Conclusions

Liver transplant is a potentially curative option for patients who present with HCC beyond the Milan criteria. In additional to LRT, ICIs have recently emerged as a potential option for either bridging or downstaging therapy for those with intermediate- or advanced-stage HCC. The concern of an increasing risk of rejection associated with ICIs prior to transplant is valid; however, several case studies have demonstrated that this approach can be executed safely. A washout period between the last dose of ICIs and liver transplantation is ideal but may not be necessary, and a minimal washout period based on drug serum half-life can be problematic, especially since the PD-1 occupancy remains high even after 4–5 half-lives. Overall, the use of ICIs in a neoadjuvant setting is of significant interest to the field of HCC and transplant oncology. Clinical trials (NCT 04,035,876 and NCT 04425226) [77,78] are currently underway and will provide some insights on the safety profile and efficacy of ICI use in this setting.

## Figures and Tables

**Table 1 cancers-13-06307-t001:** Summary of characteristics and outcomes in patients who received ICIs as bridging and/or downstaging therapy prior to liver transplant.

Authors	Age/Sex	Underlying Liver Disease	Milan Criteria	ICI	Duration of ICI	Washout Period Allowed	Donor Characteristics	Duration of Follow-Up	Complication	Rejection	Recurrence
Nordness, M.F. et al., 2020. *Am J Transplant*. [46]	65/M	HCV	Within	Nivolumab	~2 years	8 days	HCV antibody-positive, NAT-negative	Death POD 10	Hepatic necrosis	N/A	N/A
Schwacha-Eipper, B. et al., 2020. *Hapatology*. [47]	62/M	Alcoholic liver cirrhosis	Within	Nivolumab	34 cycles	15 weeks	N/A	1 year	None	None	None
Chen, G.H. et al., 2021. *Transplant Immunology*. [48]	39/M	HBV	Within	Toripalimab	10 cycles	93 days	N/A	Death POD 3	Hepatic necrosis	N/A	N/A
Sogbe, M. et al., 2021. *Transplantation.* [50]	61/M	HBV	Within	Durvalumab	15 months	>90 days	N/A	2 years	None	None	None
Tabrizian, P. et al., 2021. *Am J Transplant.* [51]	69/M	None	beyond, within UCSF criteria	Nivolumab	21 cycles	18 days	Living donor	23 months	None	None	None
	56/F	HCV	beyond, within UCSF criteria	Nivolumab	8 cycles	22 days	N/A	22 months	None	None	None
	58/M	HBV	Within	Nivolumab	32 cycles	1 days	30 U intra-op transfusion	22 months	None	None	None
	63/M	HCV, HIV	Within	Nivolumab	4 cycles	2 days	15 U intra-op transfusion	21 months	None	None	None
	30/M	HBV	Within	Nivolumab	25 cycles	22 days	N/A	16 months	None	Mild rejection	None
	63/M	HBV, HIV	Within	Nivolumab	4 cycles	13 days	N/A	14 months	Bile leak	None	None
	66/M	HBV	Within	Nivolumab	9 cycles	253 days	N/A	14 months	None	None	None
	55/F	HBV	within	Nivolumab	12 cycles	7 days	N/A	8 months	None	None	None
	53/F	NASH	beyond, within UCSF criteria	Nivolumab	2 cycles	30 days	N/A	8 months	None	None	None
Qiao, Z. et al., 2021. *Front Immunol*. [49]	7 total. All male, age 53 ± 12.1	N/A	N/A	Pembrolizumab or camrelizumab	N/A	1.3 months on average	N/A	N/A	N/A	1/7 had mild acute rejection	N/A

N/A: not applicable. NAT: nucleic acid test. POD: postoperative day.

**Table 2 cancers-13-06307-t002:** Summary of major ICIs and their associated mechanisms of action and half-lives.

Immune Checkpoint Inhibitor	Mechanism of Action	Half-Life	Reference
Ipilimumab	anti CTLA-4	14.7 days	[55]
Tremelimumab	anti CTLA-4	19.6 days	[56]
Nivolumab	anti PD-1	25 days	[57]
Pembrolizumab	anti PD-1	14–27.3 days	[58]
Cemiplimab	anti PD-1	12 days	[59]
Toripalimab	anti PD-1	14.2 days	[60]
Sintilimab	anti PD-1	35.6 h	[61]
Avelumab	anti PD-L1	3.9–4.1 days	[62]
Durvalumab	anti PD-L1	21 days	[63]
Atezolizumab	anti PD-L1	27 days	[64]

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
