# Peer review of "Liver Transplantation for Hepatocellular Carcinoma after Downstaging or Bridging Therapy with Immune Checkpoint Inhibitors"

_cancers, 2021, doi:10.3390/cancers13246307_

Round 1

Reviewer 1 Report

In this review, the Authors presented a summary of the available literature on clinical experience of using immune check point inhibitors in the neoadjuvant setting, as either bridging or downstaging therapy, for HCC before liver transplantation, that represents a new approach.

It has been already reported that ICI use may increase the risk of acute rejection and liver injury.

The Authors summarized six case reports on 20 patients, in whom ICIs were used as monotherapy, in 55% of patients Nivolumab), or combined therapy (Pebrolizumab, Camrelizumab, Toripalimab or Durvalumab), with a wide range of duration of treatment (6 weeks - 2 years), and also a variable washout period between the last dose of ICI and liver transplant (1 to 253 days).

Four studies reported successful transplant after ICI use, whereas poor outcomes were described in two studies since both patients died from liver failure early after transplant.

My comments:

- the Authors reported also two case studies on the use the neoadjuvant use of TACE with Tislelizumab or Camrelizumab in HCC patients before resection, I am not sure the patients were then transplanted, meaning that the Authors should select only cases who underwent liver transplantation after ICIs therapy

- linked to the previous comment, the Authors widely described current bridging and downstaging strategies before liver transplantation, and that chapter appears to be too long, in my opinion. The Authors should decide if the wish to discuss also on the role of ICIs before resection (without transplantation) and describe the role of ICIs in HCC before transplantation or the would like to report the other downstaging/bridging therapies. In that case, they should modify the title of the manuscript or adapt the text to the current title.

- the Authors did not report the risk of drug induced liver injury (DILI) by using ICIs. They should add a paragraph on that topic. In addition, they should make a distinction between DILI in normal liver and potential toxic effect in cirrhotic liver.

- since very few information are provided on the future potential use of ICI in the post-transplant setting, it would be of interest to understand what the Authors think on this topic.

Author Response

In this review, the Authors presented a summary of the available literature on clinical experience of using immune check point inhibitors in the neoadjuvant setting, as either bridging or downstaging therapy, for HCC before liver transplantation, that represents a new approach.

It has been already reported that ICI use may increase the risk of acute rejection and liver injury.

The Authors summarized six case reports on 20 patients, in whom ICIs were used as monotherapy, in 55% of patients Nivolumab), or combined therapy (Pebrolizumab, Camrelizumab, Toripalimab or Durvalumab), with a wide range of duration of treatment (6 weeks - 2 years), and also a variable washout period between the last dose of ICI and liver transplant (1 to 253 days).

Four studies reported successful transplant after ICI use, whereas poor outcomes were described in two studies since both patients died from liver failure early after transplant.

My comments:

- the Authors reported also two case studies on the use the neoadjuvant use of TACE with Tislelizumab or Camrelizumab in HCC patients before resection, I am not sure the patients were then transplanted, meaning that the Authors should select only cases who underwent liver transplantation after ICIs therapy

We described the use of various bridging and downstaging strategies prior to liver transplantation in section 2. Reviewer was referring to the neoadjuvant use of TACE with Tislelizumab or Camrelizumab before resection in section 3 paragraph 2. In section 3, we reviewed the current evidence with ICI use in HCC in general, NOT in the context of liver transplant; while section 4 focused on ICI use prior to liver transplant specifically. To avoid confusion, we have reworded the title of section 4.

- linked to the previous comment, the Authors widely described current bridging and downstaging strategies before liver transplantation, and that chapter appears to be too long, in my opinion. The Authors should decide if the wish to discuss also on the role of ICIs before resection (without transplantation) and describe the role of ICIs in HCC before transplantation or the would like to report the other downstaging/bridging therapies. In that case, they should modify the title of the manuscript or adapt the text to the current title.

We agree with the reviewer that there is a role of ICIs in HCC prior to resection and there are some studies on this topic in the literature1. In this review, we, however, would like to focus on the combination of ICIs and transplantation.

- the Authors did not report the risk of drug induced liver injury (DILI) by using ICIs. They should add a paragraph on that topic. In addition, they should make a distinction between DILI in normal liver and potential toxic effect in cirrhotic liver.

We appreciate this excellent suggestion and have included a paragraph describing ICI-induced hepatotoxicity at the end of section 3.

- since very few information are provided on the future potential use of ICI in the post-transplant setting, it would be of interest to understand what the Authors think on this topic.

The use of ICI in patients post transplant is a very interesting and clinically relevant topic, especially given that the risk of developing cancer is higher among solid organ transplant recipients. This is however beyond the scope of our review. A recent review by Yin et al summarized outcomes on 28 Liver transplant recipients who received ICIs2. 18 received ICIs for HCC recurrence and the rest for melanoma, CRC and SCC. 32% of those experienced biopsy-proven acute graft rejection.

1.         Pinato DJ, Cortellini A, Sukumaran A, et al. PRIME-HCC: phase Ib study of neoadjuvant ipilimumab and nivolumab prior to liver resection for hepatocellular carcinoma. BMC Cancer. Mar 23 2021;21(1):301. doi:10.1186/s12885-021-08033-x

2.         Yin C, Baba T, He AR, Smith C. Immune checkpoint inhibitors in liver transplant recipients - a review of current literature. Hepatoma Research. 2021;7:52. doi:10.20517/2394-5079.2021.11

Reviewer 2 Report

In their review paper entitled "Liver Transplantation for Hepatocellular Carcinoma after Downstaging or Bridging Therapy with Immune Checkpoint Inhibitors" Gao and coll. reported the current evidence on the application ICI in liver transplant setting.

Despite the available literature is relatively small, such field is of particular interest in transplant oncology, and will find a selected application in the next future.

The paper is generally well organized, providing a rational and concise description of the major issues of ICI application in LT setting.

In the introduction section, the Authors referred to the current allocation and prioritization policy of HCC patients in the USA: I would request them to add a brief description on the radiological definition of tumor burden following the OPTN/UNOS algorithm, that has been recently challenged by the application of LI-RADS protocol (10.1111/tri.13983) in LT setting.

Congratulations for the interesting paper

Best regards

Author Response

In their review paper entitled "Liver Transplantation for Hepatocellular Carcinoma after Downstaging or Bridging Therapy with Immune Checkpoint Inhibitors" Gao and coll. reported the current evidence on the application ICI in liver transplant setting.

Despite the available literature is relatively small, such field is of particular interest in transplant oncology, and will find a selected application in the next future.

The paper is generally well organized, providing a rational and concise description of the major issues of ICI application in LT setting.

In the introduction section, the Authors referred to the current allocation and prioritization policy of HCC patients in the USA: I would request them to add a brief description on the radiological definition of tumor burden following the OPTN/UNOS algorithm, that has been recently challenged by the application of LI-RADS protocol (10.1111/tri.13983) in LT setting.

The current allocation and prioritization policy of HCC patients in the USA is beyond the scope of this review. Although we agree with the reviewer that radiological definition of tumor burden is important, it is not particularly relevant.

Congratulations for the interesting paper

Best regards

Reviewer 3 Report

The present manuscript by Gao et al. is a comprehensive review of the role of immunotherapy alone or in combination with local ablation as bridging or downstaging strategies before liver transplantation. This is a timely and controversial topic, which is of great interest to the liver scientific community. However, the current evidence, as summarized by the authors is scarce and of low quality. Therefore, a claim of caution is required. As they currently stand, the conclusions of the review are not supported by the available evidence.

The authors are kindly invited to consider the following comments:

- In the introduction it can be read as follows: “Although there is a paucity of literature in this area, we are optimistic that checkpoint inhibitors may be utilized safely prior to transplant”. I would recommend to avoid authors’ own judgement before presenting the available data on the literature.

- As discussed by the authors, the washout period before LT is one of the most controversial aspects. I agree with the authors that avoiding rejection after LT derived from residual action of immunotherapy is an issue but the wash-out would also allow to detect patients with early HCC progression after discontinuation of immunotherapy in order to exclude them from the waiting list. Please comment.

- Some patients with advanced HCC may experience spectacular and sustained radiological response after immunotherapy, even with complete radiological response. Some of these patients could be theoretically considered for LT, even if they previously had extrahepatic metastases or macrovascular invasion. However, it is unclear whether liver transplantation would offer a survival benefit over remaining under immunotherapy. I am aware that this not solved with the available evidence but at least this caveat should be included in the text as one of the questions which will need to be solved within the next years.

- The authors’ conclusions are not supported by the available evidence. The authors stated that: “ICIs have recently become a viable option for either bridging or downstaging therapies for those with intermediate or advanced stage HCC”. This is not true. The available evidence is based in uncontrolled short case series with contrasting results, which do not allow to extract solid conclusions. The authors should be more cautious. The use of ICIs as bridging or dowstaging, either alone or in combination with other therapeutic modalities should be done exclusively within well designed randomized controlled trials. Please modify the conclusion accordingly.

- Another conclusion made by the authors was: “A washout period between the last dose of ICI and liver transplantation is ideal but may not be necessary, especially if modalities such as plasmapheresis are applied”. Again, the available evidence does not support this conclusion. Isolated case reports should not motivate such strong recommendations.

Author Response

The present manuscript by Gao et al. is a comprehensive review of the role of immunotherapy alone or in combination with local ablation as bridging or downstaging strategies before liver transplantation. This is a timely and controversial topic, which is of great interest to the liver scientific community. However, the current evidence, as summarized by the authors is scarce and of low quality. Therefore, a claim of caution is required. As they currently stand, the conclusions of the review are not supported by the available evidence.

The authors are kindly invited to consider the following comments:

- In the introduction it can be read as follows: “Although there is a paucity of literature in this area, we are optimistic that checkpoint inhibitors may be utilized safely prior to transplant”. I would recommend to avoid authors’ own judgement before presenting the available data on the literature.

We have adjusted our wording in the introduction.

- As discussed by the authors, the washout period before LT is one of the most controversial aspects. I agree with the authors that avoiding rejection after LT derived from residual action of immunotherapy is an issue but the wash-out would also allow to detect patients with early HCC progression after discontinuation of immunotherapy in order to exclude them from the waiting list. Please comment.

This is a great point. We have added that to our discussion.

- Some patients with advanced HCC may experience spectacular and sustained radiological response after immunotherapy, even with complete radiological response. Some of these patients could be theoretically considered for LT, even if they previously had extrahepatic metastases or macrovascular invasion. However, it is unclear whether liver transplantation would offer a survival benefit over remaining under immunotherapy. I am aware that this not solved with the available evidence but at least this caveat should be included in the text as one of the questions which will need to be solved within the next years.

This is a valid point and we have included this in our manuscript.

- The authors’ conclusions are not supported by the available evidence. The authors stated that: “ICIs have recently become a viable option for either bridging or downstaging therapies for those with intermediate or advanced stage HCC”. This is not true. The available evidence is based in uncontrolled short case series with contrasting results, which do not allow to extract solid conclusions. The authors should be more cautious. The use of ICIs as bridging or dowstaging, either alone or in combination with other therapeutic modalities should be done exclusively within well designed randomized controlled trials. Please modify the conclusion accordingly.

We have adjusted wording in our conclusion.

- Another conclusion made by the authors was: “A washout period between the last dose of ICI and liver transplantation is ideal but may not be necessary, especially if modalities such as plasmapheresis are applied”. Again, the available evidence does not support this conclusion. Isolated case reports should not motivate such strong recommendations.

We have adjusted wording in this specific sentence.

Round 2

Reviewer 1 Report

I am sorry to say that the Authors did not properly answer to my comments.

In particular, it appears they have not made any change to the text according to the following comments:

-the Authors widely described current bridging and downstaging strategies before liver transplantation, and that chapter appears to be too long, in my opinion. The Authors should decide if the wish to discuss also on the role of ICIs before resection (without transplantation) and describe the role of ICIs in HCC before transplantation or they would like to report the other downstaging/bridging therapies. In that case, they should modify the title of the manuscript or adapt the text to the current title.

We agree with the reviewer that there is a role of ICIs in HCC prior to resection and there are some studies on this topic in the literature1. In this review, we, however, would like to focus on the combination of ICIs and transplantation.

-since very few information are provided on the future use of ICI in the post-transplant setting, it would be of interest to understand what the Authors think on this topic

The use of ICI in patients post transplant is a very interesting and clinically relevant topic, especially given that the risk of developing cancer is higher among solid organ transplant recipients. This is however beyond the scope of our review.

Therefore, my recommendation does not change.

Author Response

I am sorry to say that the Authors did not properly answer to my comments.

In particular, it appears they have not made any change to the text according to the following comments:

-the Authors widely described current bridging and downstaging strategies before liver transplantation, and that chapter appears to be too long, in my opinion. The Authors should decide if the wish to discuss also on the role of ICIs before resection (without transplantation) and describe the role of ICIs in HCC before transplantation or they would like to report the other downstaging/bridging therapies. In that case, they should modify the title of the manuscript or adapt the text to the current title.

1st round response: We agree with the reviewer that there is a role of ICIs in HCC prior to resection and there are some studies on this topic in the literature1. In this review, we, however, would like to focus on the combination of ICIs and transplantation.

2nd round response: We appreciate the reviewer’s suggestions. We have shortened the 2nd section (current bridging and downstaging strategies prior to liver transplantation) as suggested. We have also added a paragraph on the role of ICIs in HCC before resection (without transplantation) at the end of section 4.

-since very few information are provided on the future use of ICI in the post-transplant setting, it would be of interest to understand what the Authors think on this topic

1st round response: The use of ICI in patients post transplant is a very interesting and clinically relevant topic, especially given that the risk of developing cancer is higher among solid organ transplant recipients. This is however beyond the scope of our review.

2nd round response: Although not the focus of our review, we have added a paragraph in section 4, detailing the information available on use of ICI in post transplant setting.

Reviewer 3 Report

The authors have successfully addressed my initial concerns. I think that the new version is more realistic and mirrors more accurately the available evidence. 

Author Response

We appreciate the reviewer's suggestion. 

Round 3

Reviewer 1 Report

I have not further comments